# The Natural History of Esophageal “Absent Contractility” and Its Relationship with Rheumatologic Diseases: A Multi-Center Case–Control Study

**DOI:** 10.3390/jcm11133922

**Published:** 2022-07-05

**Authors:** Daniel L. Cohen, Ram Dickman, Anton Bermont, Vered Richter, Haim Shirin, Amir Mari

**Affiliations:** 1The Gonczarowski Family Institute of Gastroenterology and Liver Diseases, Shamir (Assaf Harofeh) Medical Center, Zerifin 7030000, Israel; bermont@doctor.com (A.B.); richterv@gmail.com (V.R.); haimsh@shamir.gov.il (H.S.); 2Division of Gastroenterology, Beilinson Hospital, Rabin Medical Center, Petach Tikva 4941492, Israel; dickmanr1@gmail.com; 3Gastroenterology and Endoscopy Unit, Nazareth EMMS Hospital, Nazareth 16100, Israel; amir.mari@hotmail.com; 4Faculty of Medicine, Bar Ilan University, Safed 1311502, Israel

**Keywords:** esophageal motility disorders, scleroderma, dysphagia, achalasia, deglutition disorders

## Abstract

(1) Background: Absent contractility (AC) is an esophageal motility disorder defined as a normal integrated relaxation pressure with 100% failed peristalsis. We sought to clarify the natural history of this disorder and its relationship with rheumatologic diseases, such as systemic sclerosis (scleroderma). (2) Methods: We retrospectively identified patients with AC based on high-resolution manometry findings at three referral institutions and then matched them with controls with esophageal complaints who had normal manometries. (3) Results: Seventy-four patients with AC were included (mean age 56 years; 69% female). Sixteen patients (21.6%) had a rheumatologic disease. Compared to controls, patients with AC were significantly more likely to present with heartburn, dysphagia, vomiting, and weight loss. During follow-up, they were also more likely to be seen by a gastroenterologist, be diagnosed with gastroesophageal reflux disease, take a proton pump inhibitor, and undergo repeat upper endoscopies. No AC patients developed a new rheumatologic disease during follow-up. No significant differences were noted in the clinical presentation or course of AC patients with rheumatologic disease compared to those without. (4) Conclusions: Patients with AC have more esophageal symptoms and require more intense gastrointestinal follow-up than controls. Only a minority of patients with AC have underlying rheumatologic disease. Those without rheumatologic disease at baseline did not subsequently develop one, suggesting that a rheumatologic evaluation is likely unnecessary. The clinical course of AC in patients with rheumatologic disease and those without appears to be similar.

## 1. Introduction

Absent contractility (AC) is a disorder of esophageal peristalsis. According to the Chicago Classification, it is defined as a normal integrated relaxation pressure (IRP) with 100% failed peristalsis [1,2]. AC is the “descendant” of what was previously referred to as “scleroderma esophagus” in conventional manometry [3]. Recent studies have shown a prevalence of AC of 5–7.1% in patients with non-obstructive dysphagia undergoing esophageal high-resolution manometry (HRM) [4,5].

Numerous studies of patients with systemic sclerosis (scleroderma) have shown that many have esophageal dysmotility and that AC is the most commonly diagnosed disorder [6,7,8,9,10,11]. However, there is a dearth of literature specifically addressing patients with AC, regardless of the rheumatologic disease status. In one study evaluating the rheumatologic background of patients with AC, it was found that 63% had systemic sclerosis, while 20% had another rheumatologic disease [12]. Given the high prevalence of rheumatologic diseases, the authors recommended that all newly diagnosed AC cases be evaluated for rheumatologic disease. However, there appeared to be an element of referral bias in this study, as many patients were sent directly from a rheumatology clinic. In comparison, two more recent studies of patients with AC have shown only 37.3% and 36.1% to have an autoimmune/rheumatologic disease [13,14].

Additionally, none of these studies had a control group to compare the clinical presentation of the patients, nor did they have any follow-up. Therefore, the clinical course of AC has not been well-described, and many questions remain regarding the relationship between AC and rheumatologic diseases.

We therefore aimed to describe the natural history of AC in terms of patients’ need for gastrointestinal management, endoscopy procedures, medication use, and mortality. We also sought to evaluate the prevalence of rheumatologic diseases, such as systemic sclerosis, in patients being diagnosed with AC and to determine how many AC patients are subsequently diagnosed with rheumatologic disorders. Finally, we sought to evaluate if there are any differences in the clinical presentation or clinical course of AC patients with rheumatologic diseases versus those without.

## 2. Materials and Methods

This study was performed as a collaboration between three tertiary referral medical centers in Israel. It was approved by each center’s Institutional Review Board.

### 2.1. Patients with Absent Contractility

Manometry reports of adult patients (18 years of age and older) who underwent esophageal HRM between 2007 and 2021 were retrospectively reviewed for potential cases of AC. As some of these HRM studies were performed before the term AC was coined, the manometry reports were reviewed for several different keywords suggesting possible AC: scleroderma esophagus, scleroderma-like esophagus, absent peristalsis, absent contractility, hypocontractile esophagus, and hypotensive lower esophageal sphincter (LES).

Patients with suspected AC based on their manometry report then had their HRM data reviewed and re-analyzed by an expert in esophageal motility to confirm the correct diagnosis. AC was defined according to the Chicago Classification, version 4.0: 100% failed peristalsis with a normal median IRP value [2]. Manometries that did not meet the criteria for AC were excluded.

Given the possibility that patients with AC and a borderline IRP value may have achalasia, cases with an IRP between 10 and 15 mm Hg were excluded as the guidelines recommend and as had been done elsewhere [1,2,12].

### 2.2. Control Group

As a comparison group, patients who underwent HRM for esophageal symptoms and were found to have completely normal motility were identified. Controls were matched 1:1 to cases based on gender and age. As with the cases, the HRM data were reviewed and re-interpreted according to the Chicago Classification, version 4.0, to confirm normal manometry results [2].

### 2.3. HRM Protocol

HRM studies were performed using the ManoScan system (Medtronic, Minneapolis, MN, USA). All manometries were performed according to the standard protocol at our institutions in which 10 wet swallows were completed after successful placement of the catheter beyond the esophagogastric junction. All HRM were analyzed by senior gastroenterologists who are experts in the field of Neurogastroenterology.

### 2.4. Data Collection

Follow-up data on cases and controls were obtained through our institutions’ electronic medical records. Additionally, through a computer program, follow-up data from other hospitals and clinics throughout Israel were available. Demographic and clinical data—including ongoing gastrointestinal symptoms, repeat upper endoscopies, medication use, visits to a gastroenterology clinic or emergency room, and hospital admissions—were obtained. Any patient for whom no follow-up data were available was excluded.

### 2.5. Statistical Analysis

Statistical analysis was performed using IBM SPSS Statistics for Windows (IBM Corporation, Armonk, NY, USA). For continuous variables, *t*-tests were performed. For categorical variables, Pearson chi-square tests were used; however, for cases in which the number of variables was low, then Fischer’s exact test was utilized. For situations in which the variables were not normally distributed, the Mann–Whitney U-test was performed. For all statistical calculations, a *p*-value of <0.05 was considered significant.

## 3. Results

### 3.1. Study Population

A total of 2262 HRM procedures were performed during the study period (Figure 1). Of these, 82 studies (3.6%) were confirmed to have AC (Figure 2). Two cases were excluded due to lack of follow-up data, while six were excluded due to an IRP between 10 and 15 mmHg. Thus, 74 AC patients constituted the study population.

### 3.2. Baseline Characteristics of AC Cases

The characteristics of the study population are shown in Table 1. The mean age of the AC patients was 56.1 years, with 51 (68.9%) women. In total, 16 patients (21.6%) had rheumatologic diseases, including 5 cases of systemic sclerosis, 4 cases of lupus, and 3 cases of Sjogren’s syndrome. Ten of these cases also had Raynaud’s phenomenon. Eleven patients had undergone prior gastric surgeries, all for benign disease, including seven bariatric surgeries and four fundoplications.

Common findings on baseline endoscopy were a normal esophagus (33.8%), hiatal hernia (35.1%), and reflux esophagitis (31.1%). A dilated esophagus was noted in six cases (8.1%). Most cases were referred for HRM due to dysphagia (66.2%) or heartburn (63.5%), with vomiting, weight loss, and chest pain also being frequently cited.

### 3.3. Baseline Characteristics of Controls

There were a few significant baseline differences between the study population of AC patients and the controls with normal HRM (Table 1). Controls had less alcohol use and prior gastric surgery, and they were more likely to have a normal baseline endoscopy. AC patients more frequently had rheumatologic diseases (16 patients, 21.6%) compared to controls (1 case of Sjogren’s syndrome, 1.4%, *p* < 0.001).

AC patients were also more symptomatic that controls at presentation. Prior to their HRM, AC patients were significantly more likely to suffer from heartburn, dysphagia, vomiting, and weight loss as compared to controls.

### 3.4. Follow-Up Findings of AC Cases Compared to Controls

Follow-up data on both groups are shown in Table 2. The median follow-up for AC cases was 20.5 months, while controls were followed for a median of 55.0 months. Based on several criteria, patients with AC were more likely to have ongoing gastrointestinal issues than controls. For example, they were significantly more likely to be seen by a gastroenterologist in the follow-up, more likely to be diagnosed with GERD, more likely to be receiving a proton pump inhibitor (PPI) medication, and more likely to have repeat upper endoscopies performed. No patients were admitted to the hospital or had emergency room visits attributable to AC. Mortality was similar between the groups.

Additionally, five patients (6.8%) in the AC group were subsequently diagnosed as achalasia. This occurred despite our inclusion criteria using a cutoff level of less than 10 mmHg for IRP, as has been recommended [2]. No patients in the control group were subsequently diagnosed with achalasia.

### 3.5. Development of Rheumatologic Diseases

During follow-up, no AC patients or controls developed any new rheumatologic diseases. In total, 9 of the 58 AC patients without baseline rheumatologic disease underwent some form of rheumatologic evaluation as a result of their HRM findings. This includes five patients who were evaluated by a rheumatologist and underwent laboratory testing, including antinuclear antibodies, anticentromere antibodies, and anti-Scl 70 antibodies, which were all negative. Four others were determined not to have any rheumatologic issues based on examination and similar labs performed by their primary care physicians.

### 3.6. AC Patients with Rheumatologic Diseases Compared to Those without

Finally, we compared the 58 AC patients without rheumatologic disease to the 16 AC patients with known rheumatologic disease (Table 3). AC patients with rheumatologic diseases were more likely to be female (93.8% vs. 62.1%, *p* = 0.015) and have Raynaud’s phenomenon (56.3% vs. 1.7%, *p* < 0.001). Other baseline characteristics were similar between the groups.

During follow-up, both groups were also found to be similar in terms of their need for follow-up with a gastroenterologist, GERD diagnosis, PPI use, and repeat upper endoscopy evaluations (Table 4). No patients in either group had a new rheumatologic diagnosis during follow-up. All five cases in which AC patients were subsequently diagnosed with achalasia occurred among those without underlying rheumatologic disease. Mortality was similar between the groups.

## 4. Discussion

AC is a relatively new diagnosis, and as such, its clinical course and prognosis have not been well described. The term “absent contractility” only dates back to the Chicago Classification, version 3.0, which was published in 2015 [1]. In that classification, the authors specifically state that they changed the name of the diagnosis from “absent peristalsis,” the term used in the first two versions (2009 and 2012) of the Chicago Classification [15,16], “to differentiate the entity from other clinical scenarios in which peristalsis is absent (e.g., achalasia) [1].”

While AC, and before it “absent peristalsis,” is the name given to this condition on HRM, it is the descendant of what was often referred to as “scleroderma esophagus” on conventional manometry [3]. It was well known that abnormal manometric findings were associated with scleroderma, and they were in no way specific to it. Therefore, other names, such as “scleroderma-like esophagus” or “ineffective esophageal motility”, were also used [3].

Esophageal dysmotility occurs in up to 80–90% of patients with systemic sclerosis [6,7]. Studies using HRM have given us a better understanding of the types of esophageal motility disorders that are found among systemic sclerosis patients. For example, in a study of 79 such patients, 51% were found to have AC, 19% had ineffective esophageal motility (IEM), 6% had other major motility diagnoses, while 24% had normal motility [8]. Similar findings were seen in a study of 122 systemic sclerosis patients (60% AC, 18% IEM, 3% other diagnoses, 19% normal motility) [9]. Additionally, studies of systemic sclerosis patients have assessed the relationship between HRM findings and esophageal symptoms, as well as the relationship between HRM abnormalities and rheumatologic characteristics, such as lung involvement, skin thickness, and autoantibody profiles [11].

In contrast to the abundance of HRM studies focusing on patients with systemic sclerosis, there is a dearth of literature on patients specifically with AC. The main study evaluating patients with AC was a retrospective analysis of 207 AC patients, which found that 63% had systemic sclerosis, while 20% had another rheumatologic disease [12]. Other studies have shown a much lower prevalence (37.3 and 36.1%) [13,14]. In our study, 21.6% of AC patients had a rheumatologic diagnosis, which is much more in line with these recent AC studies, confirming that only a minority of AC patients actually have rheumatologic disease.

When comparing AC patients with rheumatologic diseases to those without, we did not identify any significant differences. It therefore appears that the clinical presentation and natural history of AC does not differ depending on whether a rheumatologic disorder is present and that these do not represent two distinct diseases. Our results are consistent with an older study in which there was no significant difference between “scleroderma esophagus” patients with or without rheumatologic diseases [17].

Similar to our results, that study also found that having “scleroderma esophagus” did not predispose to developing scleroderma in the future [17]. As no AC patients in our study subsequently developed a rheumatologic disorder, this suggests that a thorough evaluation for rheumatologic diseases in newly diagnosed AC patients without rheumatologic symptoms is unnecessary and wasteful, in contrast to what has been suggested elsewhere [12].

One of the main aims of our study was to describe the clinical course of patients with AC, as this has not been done previously. During follow-up, we found that AC patients had significant esophageal issues, even when compared to controls who underwent HRM due to esophageal symptoms. AC patients required more frequent PPI use, underwent more frequent repeat upper endoscopies, and had more frequent follow-up with a gastroenterologist. These findings suggest patients with AC have more clinically significant symptoms and require closer follow-up. While these findings were more frequent in AC patients, we did not see an increase in more serious outcomes, such as hospital admissions or mortality.

We did find that 5 of our 74 (6.8%) AC patients were eventually diagnosed with achalasia, none of whom had a rheumatologic disease. The potential for patients meeting the criteria for AC to have achalasia is well known [1,2]. It is theorized that the neuronal destruction in achalasia may not present uniformly, and therefore, some patients may have more damage to the neurons in the esophageal body initially, and only later at the LES. This would allow LES function to be preserved initially, and only over time may IRP values become higher. Indeed, with regard to AC, the Chicago classification, version 4.0, states that achalasia should be considered when IRP values are borderline (10–15 mmHg). It also suggests further investigation with a timed barium esophagram (TBE) or functional luminal imaging probe (FLIP) if dysphagia is the predominant symptom [2]. Unfortunately, none of the subjects in our study had TBE or FLIP performed.

While this study describes the natural history of AC, there are some limitations. First, this was a retrospective chart review analysis, and it is possible that some data were not available or incorrectly reported. Additionally, some patients were diagnosed with an esophageal motility disorder before the actual term “absent contractility” was coined. It is unclear if being diagnosed with “scleroderma esophagus” or “absent peristalsis”, for example, may have affected a patient’s subsequent treatment and clinical course as compared to being diagnosed with AC (information bias). Moreover, we had a relatively small sample size and length of follow-up. Finally, our control group consisting of patients with esophageal complaints who had normal HRM studies was not a unique group with a clear underlying diagnosis, and therefore, comparisons to other groups may be limited.

## 5. Conclusions

In conclusion, this is the first study to evaluate the clinical course of patients with esophageal AC. We found that AC patients frequently need follow-up and medication for their esophageal complaints, suggesting that their symptoms remain bothersome. We also found that only a small percentage of AC patients have a rheumatologic disease, such as systemic sclerosis. Since those without a rheumatologic disease did not subsequently develop one, a thorough rheumatologic evaluation of newly diagnosed AC patients is most likely unnecessary. Finally, we did not find any difference in the clinical course of those with a rheumatologic disease compared to those without. While these findings help better describe the natural history of AC and its relationship with rheumatologic diseases, future studies on patients with this fascinating diagnosis are certainly warranted.

## Figures and Tables

**Figure 1 jcm-11-03922-f001:**
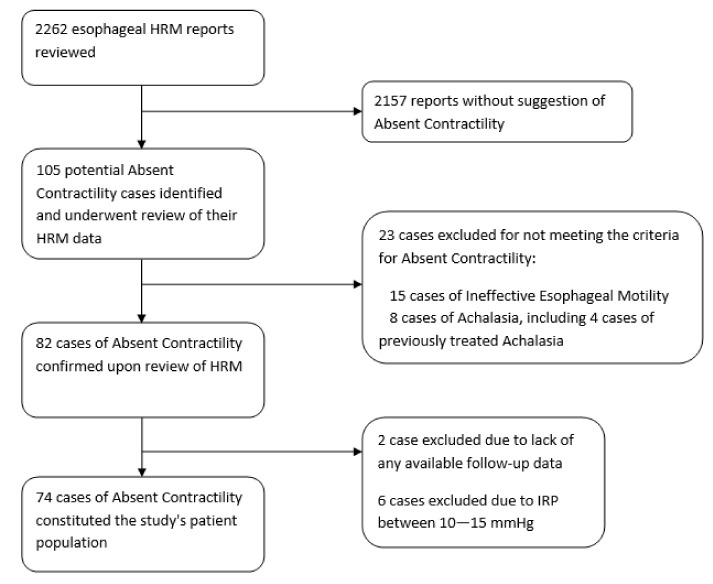
Flowchart of absent contractility patients.

**Figure 2 jcm-11-03922-f002:**
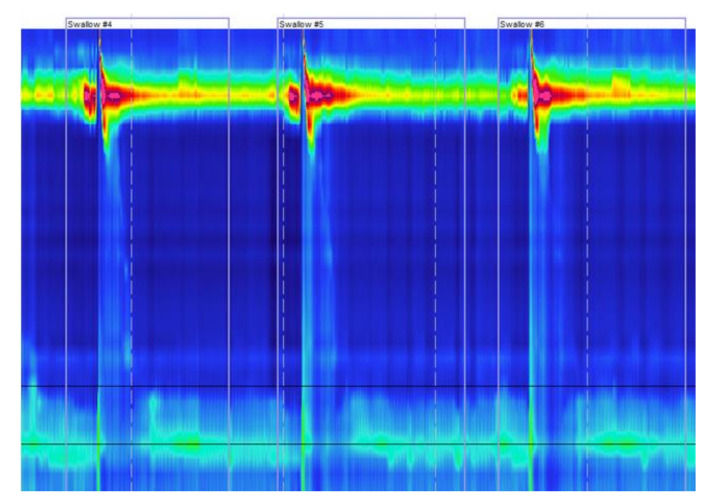
Typical high-resolution manometry findings in a patient with absent contractility. This figure shows three swallows from a 20-year-old man. No contractility is noted in the esophageal body during any of the swallows, while the lower esophageal sphincter relaxes appropriately each time.

**Table 1 jcm-11-03922-t001:** Baseline characteristics of absent contractility patients versus controls.

	AC Patients	Controls	* p * -Value
	*n* = 74	*n =* 74	
Demographics			
Age (years, SD)	56.1 (16.5)	56.2 (16.1)	0.837
Female gender	51 (68.9%)	51 (68.9%)	1.000
Pre-existing medical conditions			
Alcohol use	7 (9.5%)	0 (0%)	**0.007**
Tobacco use	24 (32.4%)	15 (20.3%)	0.093
Diabetes	12 (16.2%)	5 (6.8%)	0.071
Hypothyroidism	8 (10.8%)	4 (5.4%)	0.228
Rheumatologic disease	16 (21.6%)	1 (1.4%)	**<0.001**
Raynaud’s phenomenon	10 (13.5%)	0 (0%)	**0.001**
Prior gastric surgery	11 (14.9%)	1 (1.4%)	**0.003**
Baseline EGD findings	*n =* 68	*n =* 58	
Normal	25 (33.8%)	34 (45.9%)	**0.014**
Hiatal hernia	26 (35.1%)	13 (17.6%)	0.056
Reflux esophagitis	23 (31.1%)	8 (10.8%)	**0.009**
Barrett’s esophagus	3 (4.1%)	1 (1.4%)	0.391
Candida esophagitis	3 (4.1%)	1 (1.4%)	0.391
Retained food	3 (4.1%)	0 (0%)	0.108
Dilated esophagus	6 (8.1%)	0 (0%)	**0.004**
Epiphrenic diverticulum	2 (2.7%)	1 (1.4%)	0.655
HRM Indication			
Heartburn	47 (63.5%)	29 (39.2%)	**0.003**
Dysphagia	49 (66.2%)	30 (40.5%)	**0.002**
Vomiting	17 (23.0%)	0 (0%)	**<0.001**
Weight loss	18 (24.3%)	1 (1.4%)	**<0.001**
Chest pain	13 (17.6%)	13 (17.6%)	1.000

AC: absent contractility; SD: standard deviation; EGD: esophagogastroduodenoscopy; HRM: high-resolution manometry; *p*-values in bold represent statistically significant differences.

**Table 2 jcm-11-03922-t002:** Follow-up of absent contractility patients versus controls.

	AC Patients	Controls	*p*-Value
	*n =* 74	*n =* 74	
Length of follow-up (median months, IQR)	20.5 (8.0–43.2)	55.0 (42.0–103.2)	**<0.001**
Follow-up with a gastroenterologist	65 (87.8%)	42 (56.8%)	**<0.001**
# EGD’s performed/patient (SD)	0.68 (0.86)	0.47 (0.73)	0.374
# EGD’s performed/pt/10-year	4.48 (9.19)	0.96 (1.50)	**0.026**
Hospital admissions due to AC	0 (0%)	0 (0%)	NA
PPI use	59 (79.7%)	34 (45.9%)	**<0.001**
GERD diagnosis	53 (71.6%)	30 (40.5%)	**<0.001**
Achalasia diagnosis	5 (6.8%)	0 (0%)	**0.023**
New rheumatologic diagnosis	0 (0%)	0 (0%)	NA
Death	5 (6.8%)	3 (4.1%)	0.467

AC: absent contractility; EGD: esophagogastroduodenoscopy; GERD: gastroesophageal reflux disease; IQR: inter-quartile range; PPI: proton pump inhibitor; NA: not applicable. *p*-values in bold represent statistically significant differences.

**Table 3 jcm-11-03922-t003:** Baseline characteristics of absent contractility patients with rheumatologic disease versus those without.

	with Rheumatologic Disease	without Rheumatologic Disease	*p*-Value
	*n =* 16	*n =* 58	
Demographics			
Age (years, SD)	56.0 (16.0)	56.1 (16.8)	0.828
Female gender	15 (93.8%)	36 (62.1%)	**0.015**
Pre-existing medical conditions			
Alcohol use	0 (0%)	7 (12.1%)	0.144
Tobacco use	4 (25.0%)	20 (34.5%)	0.473
Diabetes	2 (12.5%)	10 (17.2%)	0.649
Hypothyroidism	0 (0%)	8 (13.8%)	0.116
Rheumatologic disease	16 (100%)	0 (0%)	**<0.001**
Raynaud’s phenomenon	9 (56.3%)	1 (1.7%)	**<0.001**
Prior gastric surgery	1 (6.3%)	10 (17.2%)	0.274
Baseline EGD findings	*n =* 14	*n =* 54	
Normal	5 (35.7%)	20 (37.0%)	0.927
Hiatal hernia	4 (28.6%)	22 (40.7%)	0.404
Reflux esophagitis	7 (50.0%)	16 (29.6%)	0.151
Barrett’s esophagus	0 (0%)	3 (5.6%)	0.367
Candida esophagitis	1 (7.1%)	2 (3.7%)	0.577
Retained food	0 (0%)	3 (5.6%)	0.367
Dilated esophagus	3 (21.4%)	6 (11.1%)	0.310
Epiphrenic diverticulum	0 (0%)	2 (3.7%)	0.465
HRM Indication			
Heartburn	12 (75.0%)	35 (60.3%)	0.281
Dysphagia	10 (62.5%)	39 (67.2%)	0.723
Vomiting	4 (25.0%)	13 (22.4%)	0.828
Weight loss	3 (18.8%)	15 (25.9%)	0.557
Chest pain	1 (6.3%)	12 (20.7%)	0.179

SD: standard deviation; EGD: esophagogastroduodenoscopy; HRM: high-resolution manometry; *p*-values in bold represent statistically significant differences.

**Table 4 jcm-11-03922-t004:** Follow-up of absent contractility patients with rheumatologic disease versus those without.

	with Rheumatologic Disease	without Rheumatologic Disease	*p*-Value
	*n =* 16	*n =* 58	
Length of follow-up (median months, IQR)	21.5 (9.0–89.7)	20.0 (7.7–50.5)	0.490
Follow-up with a gastroenterologist	4 (80.0%)	18 (94.7%)	0.963
# EGD’s performed/patient (SD)	0.81 (0.98)	0.64 (0.83)	0.081
# EGD’s performed/pt/10-year (SD)	3.48 (6.17)	4.76 (9.89)	0.927
Hospital admissions for AC	0 (0%)	0 (0%)	1.000
PPI use	13 (81.3%)	46 (79.3%)	0.864
GERD diagnosis	13 (81.3%)	40 (69.0%)	0.335
Achalasia diagnosis	0 (0%)	5 (8.6%)	0.224
New rheumatologic diagnosis	0 (0%)	0 (0%)	NA
Death	1 (6.3%)	4 (6.9%)	0.927

AC: absent contractility; SD: standard deviation; EGD: esophagogastroduodenoscopy; GERD: gastroesophageal reflux disease; IQR: inter-quartile range; PPI: proton pump inhibitor; NA: not applicable.

## Data Availability

The database is available from the corresponding author upon reasonable request.

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
