# Peer review of "The Natural History of Esophageal “Absent Contractility” and Its Relationship with Rheumatologic Diseases: A Multi-Center Case–Control Study"

_jcm, 2022, doi:10.3390/jcm11133922_

Round 1

Reviewer 1 Report

A very interesting scientific issue, rarely mentioned in the literature. The work is of great practical importance for gastroenterologists and surgeons of the upper gastrointestinal tract. Well planned and well written publication.  I am a supporter of a richer confrontation with scientific literature. And this is my only objection to the manuscript.

Author Response

We thank the reviewer for their kind words.  We agree that a richer confrontation with the scientific literature is preferred.  However, as the topic of Absent Contractility is scarcely mentioned in the literature, our Discussion was therefore a bit limited.  We did the best that we could do and cited all of the published studies on the topic.

Reviewer 2 Report

I read with interest authors' work. This is indeed an intriguing paper even if I can't detect a true novelty. It seems that there is no major concerns found in the section. Some conclusions have not been discussed or have not be shown by scientific data before. English might be improved.

Author Response

We thank the reviewer for their kind words.  We are a bit surprised by the comment on the quality of the English in the manuscript since the lead author who wrote this paper is a native English speaker, born and raised in the United States.  Perhaps the comment stems from the fact that some of the terms and spelling reflect an American English as opposed to a British English, but we are confident that the English language in the manuscript is more than up to par.

Reviewer 3 Report

In this retrospective study, the authors aimed at elucidate the natural history of absent contractility (AC) of the esophagus and its relation to rheumatologic diseases. They demonstrated that (i) patients with esophageal AC frequently need follow-up and medication for their esophageal complaints and (ii) only a small percentage of AC patients have a rheumatologic disease such as systemic sclerosis. Overall, this is a well-written paper providing interesting data. Despite its inherent limitations with retrospective design, their results can be invaluable future references for esophageal care-givers. I did not observe major flaws and have some minor comments as below.

1)    The study population of AC patients had a history of prior gastric surgery more often as compared to the control group. This is a very interesting finding and the readers would want to know more information about it. For instance, did these prior gastric surgeries include oncological gastrectomy?

2)    The authors described “The potential for patients meeting the criteria for AC to have achalasia is well-known”. This should be followed by an optimal citation.

3)    The authors stated “While these findings help better describe the natural history of AC and its relationship with rheumatologic diseases, future multi-center studies with more participants and longer durations of follow-up on patients with this fascinating diagnosis are certainly warranted”. This sentence was not necessary to be included to Conclusions.

Author Response

  1. The study population of AC patients had a history of prior gastric surgery more often as compared to the control group. This is a very interesting finding and the readers would want to know more information about it. For instance, did these prior gastric surgeries include oncological gastrectomy?

Thank you for the comment as we had neglected to fully discuss this point.  None of the prior surgeries were for oncological reasons.  The majority were bariatric procedures, as well as some fundoplications.  We have added a sentence to the Results section clearly stating this (page 4 lines 129-131).

  1. The authors described “The potential for patients meeting the criteria for AC to have achalasia is well-known”. This should be followed by an optimal citation.

We have added the relevant citations to this comment (page 9, line 252).

  1. The authors stated “While these findings help better describe the natural history of AC and its relationship with rheumatologic diseases, future multi-center studies with more participants and longer durations of follow-up on patients with this fascinating diagnosis are certainly warranted”. This sentence was not necessary to be included to Conclusions.

We have edited this sentence to make it a more appropriate concluding sentence for the Conclusions section (page 9, line 281).